# Training Convolutional Neural Networks to Score Pneumonia in Slaughtered Pigs

**DOI:** 10.3390/ani11113290

**Published:** 2021-11-17

**Authors:** Lorenzo Bonicelli, Abigail Rose Trachtman, Alfonso Rosamilia, Gaetano Liuzzo, Jasmine Hattab, Elena Mira Alcaraz, Ercole Del Negro, Stefano Vincenzi, Andrea Capobianco Dondona, Simone Calderara, Giuseppe Marruchella

**Affiliations:** 1AImageLab, University of Modena and Reggio Emilia, Via Vivarelli 10/1, 41125 Modena, Italy; lbonicelli@unimore.it (L.B.); ercole@farm4trade.com (E.D.N.); simone.calderara@unimore.it (S.C.); 2Faculty of Veterinary Medicine, University of Teramo, Loc. Piano d’Accio, 64100 Teramo, Italy; artrachtman@unite.it (A.R.T.); jhattab@unite.it (J.H.); elenamiraalcaraz@gmail.com (E.M.A.); 3Department of Veterinary Public Health, Azienda Unità Sanitaria Locale di Modena, via S. Giovanni del Cantone 23, 41121 Modena, Italy; a.rosamilia@ausl.mo.it (A.R.); g.liuzzo@ausl.mo.it (G.L.); 4Farm4Trades.r.l., Via IV Novembre, 66041 Atessa, Italy; s.vincenzi@farm4trade.com (S.V.); andrea@farm4trade.com (A.C.D.)

**Keywords:** pig, slaughterhouse, pneumonia, scoring methods, artificial intelligence, deep learning, convolutional neural networks

## Abstract

**Simple Summary:**

Scoring lesions in slaughtered pigs can provide useful feedback to the swine industry, although the systematic recording of lesions is very challenging and time consuming. Artificial intelligence offers interesting opportunities to solve highly repetitive tasks, such as those performed by veterinarians at postmortem inspection in high-throughput slaughterhouses and to consistently analyze large amounts of data. The present investigation indicates that enzootic pneumonia-like lesions can be effectively detected and quantified through artificial intelligence methods under routine slaughter conditions.

**Abstract:**

The slaughterhouse can act as a valid checkpoint to estimate the prevalence and the economic impact of diseases in farm animals. At present, scoring lesions is a challenging and time-consuming activity, which is carried out by veterinarians serving the slaughter chain. Over recent years, artificial intelligence(AI) has gained traction in many fields of research, including livestock production. In particular, AI-based methods appear able to solve highly repetitive tasks and to consistently analyze large amounts of data, such as those collected by veterinarians during postmortem inspection in high-throughput slaughterhouses. The present study aims to develop an AI-based method capable of recognizing and quantifying enzootic pneumonia-like lesions on digital images captured from slaughtered pigs under routine abattoir conditions. Overall, the data indicate that the AI-based method proposed herein could properly identify and score enzootic pneumonia-like lesions without interfering with the slaughter chain routine. According to European legislation, the application of such a method avoids the handling of carcasses and organs, decreasing the risk of microbial contamination, and could provide further alternatives in the field of food hygiene.

## 1. Introduction

Respiratory syndromes are recognized worldwide as a major concern for the profitability of livestock farming. This is particularly true in the modern swine industry, where large groups of pigs are reared under confined and intensive conditions. Prolonged exposure to adverse environments (e.g., inadequate ventilation systems, inhalation of large amounts of dust and irritating chemicals, such as ammonia) overwhelms the effectiveness of respiratory defenses, thus facilitating the occurrence of infections. Moreover, the presence of dense pig populations increases the spread of pathogens and the burden of infectious diseases. As a result, pig farms may suffer substantial economic losses, because of increased mortality and costs for veterinary cares, reduced daily weight gain and feed conversion efficiency [1].

Suitable laboratory tools are currently available to demonstrate the presence/absence of pathogens in pig herds, as well as to monitor the kinetics of infection along the pig flow [2,3]. However, the occurrence and the severity of respiratory diseases result from several interrelated factors, well beyond the spread of infectious agents in a given pig population. Therefore, estimating the real impact of respiratory diseases can be challenging, mainly for chronic diseases characterized by high morbidity and low mortality. In such cases, the slaughterhouse can act as a valid and efficient checkpoint, considering that chronic lesions are still evident at postmortem inspection and provide useful information about the prevalence and the economic impact of diseases [2,4].

Enzootic pneumonia (EP) is caused by *Mycoplasma hyopneumoniae* and is still widespread in most of major swine-raising countries, where it is regarded as a key component of the “porcine respiratory disease complex” (PRDC). Enzootic pneumonia occurs in grower and finishing pigs as a chronic respiratory disease, mainly characterized by persistent, nonproductive coughing, impaired growth and feed efficiency. Secondary bacterial infections are a common event and play a substantial role in worsening clinical signs and increasing mortality rate [5,6]. Pathological findings are typical, although not pathognomonic of EP (so-called “EP-like” lesions), and consist of well-demarcated, bilateral pneumonic foci affecting the cranioventral portions of the lungs. Over time, EP-like lesions evolve from reddish to grayish areas, often surrounded and/or intermingled with emphysematous lobules [7,8]. Naturally, pneumonia tends to heal and might appear as scars (“fissures”) at slaughter, mainly due to the timing of infection and the market age of the pigs. Nevertheless, the evaluation of EP-like lesions can be conveniently carried out in slaughtered pigs, thus representing a valuable tool to estimate the prevalence and the severity of EP, as well as the efficacy of the implemented control strategies [4,9,10].

At present, EP-like lesions scoring is performed by veterinarians serving the slaughter chain and it is a costly, challenging and time-consuming activity. Different methods have been developed to score EP-like lesions [11], all of them being inspired by the same idea: the larger the lesion is, the more severe the impact is on pig growth. In Europe, “Madec’s grid” is the most widely used pneumonia scoring system, as it is quite fast and can be performed in large high-throughput slaughterhouses [12]. According to Madec’s grid, each lobe is inspected and palpated, divided into quarters and scored from 0 to 4 points regardless of its size, thus equally contributing to the final score. To solve this issue, Madec’s grid is usually combined with a method to account for each lobe volume, as proposed by Christensen et al. [2]. Several studies have shown a negative correlation between the severity of EP-like lesions at postmortem inspection and the growth performances of pigs. As an example, Morris et al. [13] observed a mean decrease in the final weight of 1.8 kg for each 10% of EP-like lesions.

Deep learning (DL) represents a very powerful tool to consistently analyze large amounts of data. Overall, DL is a broad term indicating a family of machine learning algorithms, which can extract high order features by stacking several operations in a sequence of individual blocks (so-called “layers”). Within DL, convolutional neural networks (CNNs) are widely recognized as the state of the art for computer vision, which is a subset of artificial intelligence (AI) aiming to automate the recognition of features in digital images or videos, well-adapted to solving highly repetitive visual tasks [14,15]. In a CNN, the convolution operation replaces large and dense layers with many low-dimensional “filters”, bringing the following two relevant benefits: (1) reducing the number of parameters to be trained; (2) forcing the learned features to be invariant with respect to translations in the input. These key properties limit the chance of over-fitting the training data and provide a faster training time, by decreasing the computational cost of the model [16].

Over recent years, AI has been applied to several aspects of modern life, gaining traction in many fields of research, including livestock production [17]. The present study aims to develop a CNN capable of recognizing and quantifying EP-like lesions in slaughtered pigs. Such a CNN would be complementary to other ones, already trained and currently available [18], thus allowing the systematic collection and analysis of data on the prevalence and severity of respiratory diseases in pigs.

## 2. Materials and Methods

### 2.1. Animals and Photo Collection

Investigations were carried out in three different and high-throughput abattoirs (slaughter chain speed = 6–8 pigs/min). Two slaughterhouses were located in Italy and one slaughterhouse was located in Spain. The Italian slaughterhouses were processing “heavy” pigs with an approximate slaughter weight of 160 kg and an average age of 9–10 months. The Spanish slaughterhouse was processing “light” pigs with an approximate slaughter weight of 90 kg and 5 months of age.

Pictures were taken by veterinarians along the slaughter chain under routine slaughter conditions, by means of smartphone cameras (Apple iPhone SE, Apple iPhone XS Max, Xiaomi Redmi Note 8T, Xiaomi Realme 7 pro). All veterinarians involved in the present study had skills in porcine respiratory pathology and could palpate the lungs to confirm/rule out pneumonia, if necessary. Each lung was photographed in such a way that its external surface occupied most of the field of view (Figure 1a). Lungs entirely filled with blood or severely “ripped” because of chronic pleurisy were not included in the study (Figure 1b,c).

### 2.2. Photo Annotation

A total of 7564 pictures were selected and annotated by the veterinarians (A.R.T., J.H. and G.M.) using an open-source image segmentation tool [19], after multiple sessions of training in order to agree on the way to annotate pictures. In particular, the following areas of interest were annotated: LUNG, i.e., the entire silhouette of the lung surface; LESION, i.e., pneumonic foci compatible with EP-like lesions; LOBE, i.e., the cranial lobe, which was separately annotated when it was bent over, thus partially overlapping with the middle and/or the diaphragmatic lobes (see Figure 2 for details). The veterinarians (A.R.T., J.H. and G.M.) equally contributed to the dataset and the annotated pictures were randomly assigned to the “train set” or to the “test set” (see below for details).

### 2.3. Architecture of the DL-Based Model Employed

For the task at hand, we developed a DL-based model (DLBM) to accurately predict and segment the area of lung, lesion and lobe given an input image. Specifically, we employed a convolutional auto-encoder architecture based on U-Net [20], which has been extensively modified to improve the overall segmentation performance.

The auto-encoder architecture consists of two main components, namely, the “encoder” and “decoder” networks. The encoder is responsible for producing a lower-dimensional representation of the input image, while the decoder up-scales and further processes the input image to extract the final output. In our solution, we developed the encoder after the convolutive section of the well-established ResNet34 model [21]. We pre-trained the network using established classification datasets, thus providing better parameter initialization by means of knowledge transfer to our task. The decoder network is composed of a sequence of up-sampling convolutions, each one mirroring the corresponding encoder layer. Notably, this allows for the use of skip connections between the two parts of the model. Skip or residual connections facilitate the propagation of the information through the network, reducing both the time and data required for the model to converge.

According to Zhou et al. [22], we finally enriched the representation provided by each residual path by including several intermediate convolutive blocks, with dense skip connections between each one. These layers further combine the features extracted by the encoder and facilitate the semantic shift between the encoder and decoder representations. The DLBM employed herein is graphically represented in Figure 3.

### 2.4. Training Process

According to Lin et al. [23] and Chen et al. [24], we employed an image pyramid structure and supplied the model with multiple-scaled versions of the same picture. Each input was resized through bilinear interpolation and independently pre-processed by means of a small convolutive block. In More detail, each encoder block received (a) the output from the previous block of the chain (if any) and (b) the appropriate pre-processed input. This strategy allows the encoder to more efficiently recover and combine features at different scales.

We trained the overall architecture with deep supervision [25] to produce an output at each intermediate stage of the first residual connection, thus providing additional paths for the gradient to flow. This strategy can further reduce the time required for the algorithm to converge. Finally, at inference time, we only keep the output produced by the last decoder.

The model takes one RGB image as the input and produces a binary segmentation mask for each class, each one with the same resolution as the input. This strategy makes it possible for a pixel to belong to more than one class at the same time (e.g., lung and lesion). We trained the network for a total of 400 epochs, where each epoch includes all the training samples, and augments the data with random horizontal flips, random crops, random translations and rotations and random color jitter—change in brightness, contrast, saturation and hue—to improve generalization.

### 2.5. Dataset

The entire dataset was split between “train” and “test sets”. Both sets consist of annotated pictures, which represent the ground-truth label of each pixel. The training set includes 7154 pictures; at this stage, the veterinarians’ annotations (VAs) aim to improve the performance of the DLBM. The test set consists of 410 images and was shown to the network only during the inference stage, when no weight in the network could be altered. In this case, the ground-truth is used to evaluate the performance of the model.

### 2.6. Metrics

Based on the presence/absence of EP-like lesions, input pictures were classified as diseased or healthy lungs. The performances of the DLBM were computed in terms of sensitivity (i.e., the ability to correctly identify diseased lungs) and specificity (i.e., the ability to correctly identify healthy lungs), with respect to the VAs (i.e., the gold standard).

The distance between the pluck and the camera affects the apparent size of pneumonia, as quantified by counting the number of pixels of the class “lesion”. Therefore, the size of EP-like lesions was expressed as a percentage, by computing the following ratio (1):(1)$$r=\frac{\#\left\{lesion\right\}}{\#\left\{lung\right\}+\#\left\{lobe\right\}}$$
where #{*lung*}, #{*lesion*} and #{*lobe*} represent the number of pixels of each respective class.

The quantification of EP-like lesions is directly linked to the segmentation performance of the model, as the incomplete or improper prediction of each class (“lung, lobe, lesion”) can affect the predicted ratio (r). Therefore, the Intersection over Union (IoU) for each class was additionally computed, averaging the results across the test set (see Figure 4 for details).

Formally, given a ground-truth mask *y_c_* and a predicted mask ŷ*_c_*, the IoU (*y_c_*,ŷ*_c_*) is defined as in the following Equation (2):(2)$$IoU\left(y_{c},{\hat{y}}_{c}\right)=\frac{\left|y_{c}\cap {\hat{y}}_{c}\right|}{\left|y_{c}\cup {\hat{y}}_{c}\right|}.$$

Finally, the correlation between VAs and DLBM predictions was calculated (Pearson’s coefficient).

## 3. Results

### 3.1. Training Set—Data Provided by the Veterinarians

The entire training set consisted of 7154 annotated pictures. The veterinarians identified 3283 lungs as healthy (45.89%), while EP-like lesions were detected in the remaining 3871 pictures (54.10%). The cranial lobe was annotated in 15.35% of the healthy lungs (504 pictures) and in 20.82% of the diseased lungs (806 pictures).

In the diseased lungs, the size of the lesions ranged between 0.12 and 80.19% (mean value ± SD = 9.71 ± 9.16%; median = 6.9%). The main features of the training set are graphically summarized in Figure 5.

### 3.2. Test Set—Data Provided by the Veterinarians

The whole test set consisted of 410 pictures. According to the VAs, 159 pictures proved to be healthy (38.78%), while the cranial lobe was annotated in 25 cases (15.72% of the healthy lungs). The veterinarians identified EP-like lesions in the remaining 251 images (61.21%), the cranial lobe being annotated in 74 cases (29.48% of diseased lungs). In the diseased lungs, the size of the lesions ranged between 0.07 and 63.02% (mean value ± SD = 10.47 ± 9.55%; median = 7.52). Figure 6 provides more detailed information about the size of the EP-like lesions in the diseased lungs.

### 3.3. Test Set—Data Predicted by the DL-Based Method

The data are graphically summarized in Figure 6. As additionally shown in Table 1, the DLBM employed herein showed high sensitivity and specificity values.

In more detail, the DLBM gave only one false positive prediction, due to the presence of a small lesion on the opposite lung. The DLBM overlooked three small lesions (<2% of the lung surface), always located on the apex of the middle lobe, while it correctly predicted all the lungs affected by larger EP-like lesions (Figure 7, Figure 8, Figure 9).

In the lungs considered as diseased after the veterinarians’ evaluation, the predicted size of the lesions ranged between 0% (i.e., false negative) and 61.05% (mean value ± SD = 10.52 ± 9.50%; median = 7.99).

In terms of IoU, the performances of the DLBM are reported in Table 2. Overall, the average IoU was around 0.80 for the “lobe” and “lesion” classes, while a much higher value was reached for the class lung (0.97).

The correlation rate between the veterinarians’ and DLBM scores was very high (Pearson’s coefficient = 0.96).

A demo version of the DLBM is provided as Appendix A.

## 4. Discussion

A growing body of evidence indicates that abattoir inspections are very useful to monitor the health status and the welfare of farm animals. Data collected at the slaughterhouse can offer reliable information to the entire swine industry, thus allowing for a more effective and efficient management of health and welfare issues [10,26,27,28,29]. Several Northern European countries have adopted specific schemes to record inspection data from slaughtered pigs at a national level [30,31,32,33,34,35]. Likewise, abattoir inspections have been included in a new information system (namely, ClassyFarm) in Italy, implemented by the Ministry of Health to categorize farms according to risk [36].

During the last few decades, mainly since the advent of DL, the application of AI in medicine has dramatically expanded [37]. In our opinion, AI-based methods can be successfully applied to solve highly repetitive tasks, such those performed by veterinarians to score pluck lesions in high-throughput slaughterhouses.

To date, few DLBMs have been developed to monitor abattoir lesions in pigs [18,38,39]. The DLBM employed herein demonstrated that it was able to suitably recognize lungs as healthy or diseased, showing sensitivity and specificity close to 100% when compared with the VAs. In addition, the size of the lesions in terms of mean ± SD and median, as annotated by the veterinarians or predicted by the DLBM, was demonstrated to be almost identical. Considering the IoU, the DLBM performances proved outstanding for the class “lung” and lower, although satisfactory for the classes “lobe” and “lesion”. Such differences are likely due to the following several factors: (a) the training set for the class “lung” is larger, as “lesion” and “lobe” are not present in all the pictures; (b) outlining the borders of “lobe” can be challenging and somewhat subjective; (c) the shape of the lesions can be irregular, due to the presence of fissures and/or emphysematous lobules.

Sibila et al. [40] and Garcia-Morante et al. [11] developed scoring systems to assess *Actinobacillus pleuropneumoniae* and *Mycoplasma hyopneumoniae* lesions on digital images, under experimental conditions. In more detail, these authors quantified the lesions as a percentage of the total pulmonary surface, after manually outlining each area of interest.

Such image analyses seem to correlate with other scoring systems, although lesions of the accessory lobe are not accounted for and partially affect the Pearson’s coefficient [11].As a matter of fact, the DLBM employed herein pursues a similar approach. In fact, we did not evaluate the accessory lobe, which remains hidden from the camera. Garcia-Morante et al. [11] calculated the formulae of equivalence to compare the different scoring methods. Likewise, we consider that the ad hoc formulae of equivalence could suitably solve this issue also using DLBMs. Interestingly, simplified slaughter check procedures have been proposed, based on the examination of a single lung; in this case, the final score is computed by multiplying by the correcting factors [2]. The combination of DLBMs with such simplified procedures should be tested, to make the collection of pictures easier and to further improve the efficiency of the method.

In our experience, severe artifacts due to the slaughter technique (i.e., inspiration of a large amount of blood, tearing of lungs after chronic pleurisy) are not so common as to compromise the evaluation of the entire batch. Likewise, different environments (e.g., brightness, distance between pluck and operator) and mild artifacts (e.g., pleural petechiae, small amounts of blood or foam on the lungs) do not impair the performances of the DLBM. In this respect, we highlight that a great effort was made to train the DLBM under scenarios as varied and diverse as possible, through extensive data augmentation activity. Nevertheless, some technical issues should be solved to apply the DLBM on a large scale. First, image capture should be automated. In this respect, preliminary tests with robot cameras have given promising results, in order to simultaneously automate several activities along the slaughter chain (e.g., traceability of animals through reading codes tattooed on the thigh). The standardization of the distance between the pluck and the camera could allow the scoring of pneumonia even in most ripped lungs, by counting the number of pixels of the class “lesion”. In addition, the training of the staff, along with few structural adjustments (e.g., hooks to avoid the flipping of the cranial lobe), could further improve the performance of the DLBM.

## 5. Conclusions

CNNs can successfully recognize the most economically relevant disease conditions in slaughtered pigs [18,39] and could provide large-scale data analysis, without interfering with the slaughter chain routine. The present investigation further widens the field of application of DLBMs, which now includes another major component of the PRDC. Moreover, we consider that DLBMs could offer interesting, effective and cost-efficient alternatives in the field of food safety. As a matter of fact, the European Regulation 2019/627 [41] provides the visual postmortem inspection, at least in the first instance, to avoid the handling of carcasses and to decrease the risk of microbial contamination [42].Some data already suggest that the “new technologies” allow the remote inspection at slaughter [43]. In the next future, AI-based methods could suitably meet this need, allowing a further leap in quality in the field of food inspection, supporting official veterinarians and optimizing the management of human resources, especially in high-throughput slaughterhouses.

## Figures and Tables

**Figure 1 animals-11-03290-f001:**
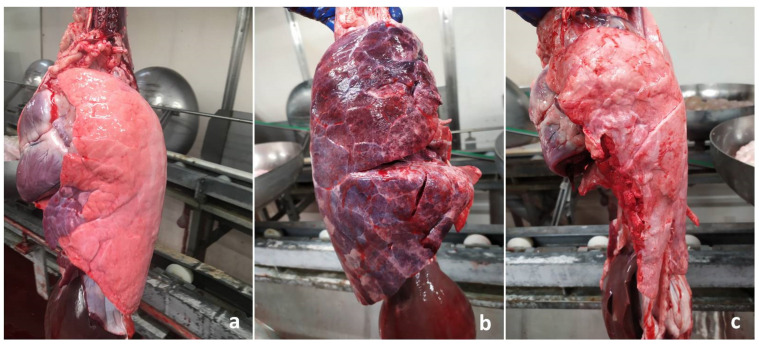
Slaughtered pigs. Lungs. (**a**) The image clearly shows the external surface of the left lung, an EP-like lesion affecting the middle lobe; this kind of picture was considered suitable for the present investigation. (**b**) The right lung has a diffuse reddish appearance, because of the inspiration of blood after exsanguination. (**c**) A large portion of the left lung is missing, as it remained attached to the chest wall because of chronic pleurisy. Both (**b**,**c**) pictures were considered unsuitable and were not included in this study.

**Figure 2 animals-11-03290-f002:**
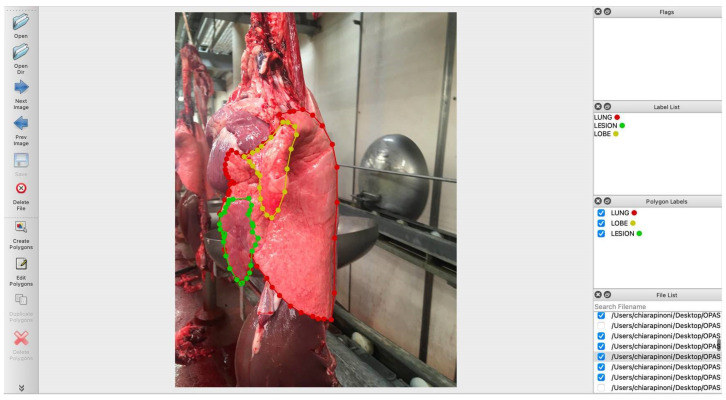
Label me annotation tool. In this case, three areas have been annotated, i.e., the silhouette of the left lung (“lung”, red dots and lines), the flipped cranial lobe (“lobe”, yellow dots and lines) and an EP-like lesion (“lesion”, green dots and lines). Both lobe and lesion are fully embedded within the “lung” borders.

**Figure 3 animals-11-03290-f003:**
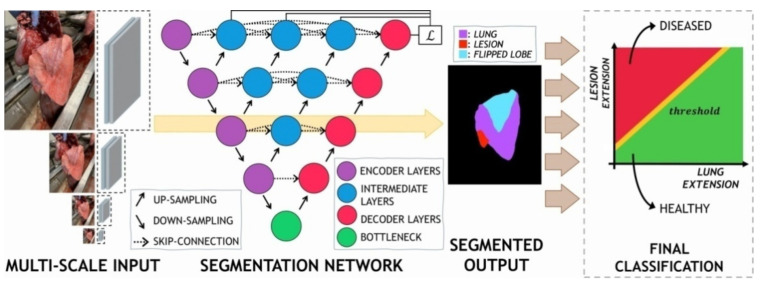
Architecture of the DL-based model employed. The input is scaled to multiple resolutions and pre-processed before being fed to the main network. The segmentation masks delivered can be conveniently depicted as an RGB image by assigning a color to each class (red, purple, blue). The final score is computed as the ratio between the number of pixels belonging to the class “lesion” and those belonging to the combined classes “lung + lobe” (i.e., the entire pulmonary size).

**Figure 4 animals-11-03290-f004:**
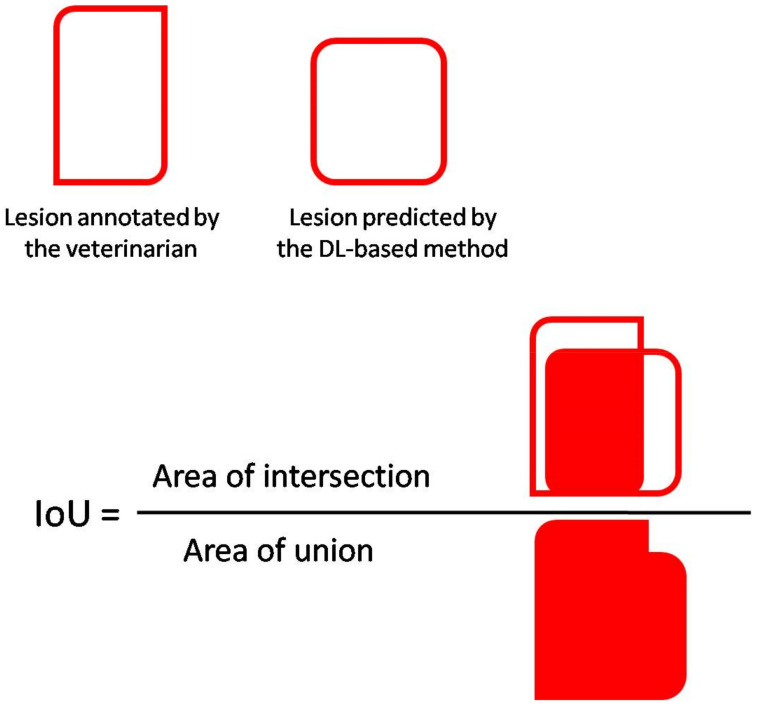
A graphical representation of the Intersection over Union (IoU). IoU can range between 0 and 1and it is recognized as a good metric for measuring overlap between two “masks”. The IoU value 1 is when the prediction is completely correct (i.e., when the prediction of the DLBM fully overlaps with the annotation of the veterinarian). Conversely, the lower the IoU is, the worse the prediction of the DLBM is (the IoU value is 0 when the two masks do not overlap at all).

**Figure 5 animals-11-03290-f005:**
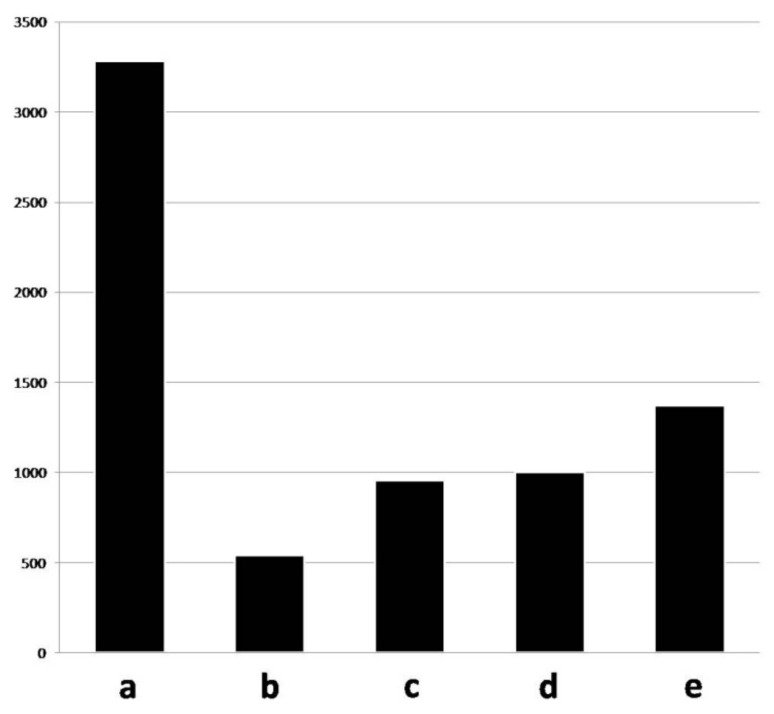
Main features of the training set. Overall,3283 lungs were recognized as healthy (**a**), while EP-like lesions were detected in the remaining 3871 pictures (**b** + **c** + **d** + **e**). The size of the lesion was <2% of the entire lung surface in 541 cases (**b**), between 2 and 5% in 958 cases (**c**), between 5 and 10% in 1001 cases (**d**) and >10% in 1371 pictures (**e**).

**Figure 6 animals-11-03290-f006:**
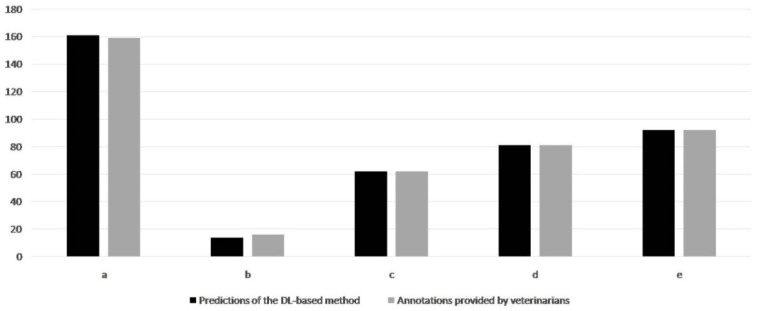
Test set. Comparison between the veterinarians’ annotations and the predictions of the DL-based method. Graphics clearly show that the DLBM predictions are very similar to the gold standard provided by the veterinarians, both for healthy and diseased lungs, regardless of the size of the EP-like lesions. Legend: (**a**) healthy lungs; (**b**) lesion size <2% of the entire lung surface; (**c**) lesion size between 2 and 5%; (**d**) lesion size between 5 and 10%; (**e**) lesion size >10%.

**Figure 7 animals-11-03290-f007:**
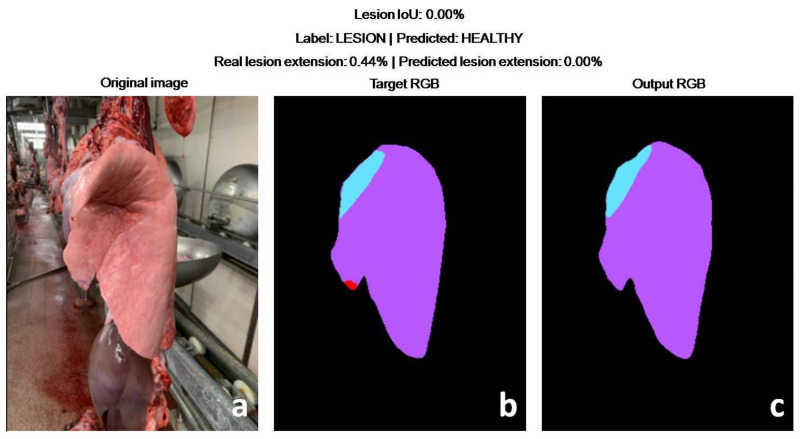
Test set. False negative prediction provided using the DL-based method. Pictures (**a**,**b**) represent the original picture and the annotation of the veterinarian, respectively. In detail, the veterinarian-annotated lung (purple), lobe (blue) and lesion (red). As shown in picture ((**c**); DLBM prediction), the DLBM overlooked the very small lesion on the tip of the middle lobe.

**Figure 8 animals-11-03290-f008:**
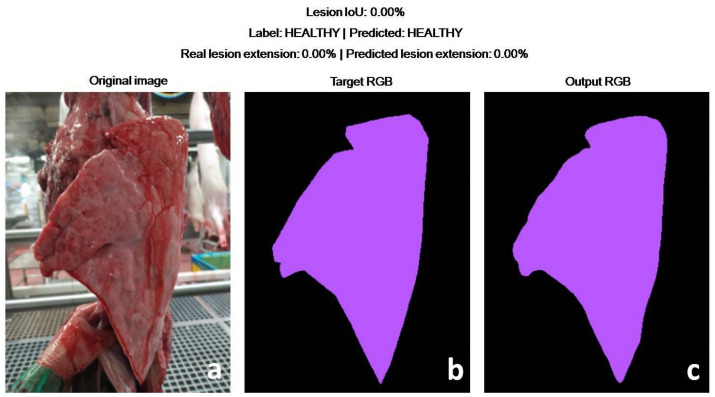
Test set. Healthy lung. Pictures (**a**,**b**) represent the original picture and the annotation of the veterinarian, respectively. In this case, the veterinarian annotated only the lung (purple).The prediction of the DLBM (**c**) perfectly matches the annotation of the veterinarian (**b**). The examination of the original picture (**a**) confirms that the lung is healthy, no lesion being detectable.

**Figure 9 animals-11-03290-f009:**
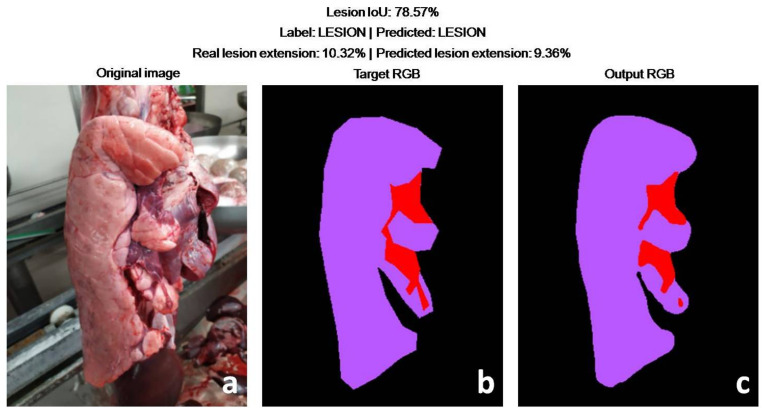
Test set. Diseased lung. Pictures (**a**,**b**) represent the original picture and the annotation of the veterinarian, respectively. In detail, the veterinarian-annotated lung (purple) and lesion (red). The prediction of the DLBM (**c**) largely, although not perfectly, fits with the VAs (**b**). As shown by the original pictures (**a**), the shape of the EP-like lesion is irregular, due to the presence of fissures and intermingling emphysematous lobules. Notably, the DLBM also well-predicted a long cut (artifact) in the diaphragmatic lobe.

**Table 1 animals-11-03290-t001:** Sensitivity and specificity of the DL-based method on the test set.

	Number of Pictures, as Interpreted and Annotated by the Veterinarians (Gold Standard)	Number of Pictures Correctly Predicted using the DL-Based Method	Sensitivity (%)	Specificity (%)
Lesion size <2% of the entire lung surface	16	13	81.25	//
Lesion size between 2 and 5% of the entire lung surface	62	62	100	//
Lesion size between 5 and 10% of the entire lung surface	81	81	100	//
Lesions >10% of the entire lung surface	92	92	100	//
Healthy lungs	159	158	//	99.38

**Table 2 animals-11-03290-t002:** Test set. Intersection over Union. Performances of the DL-based method with respect to the veterinarians’ annotations.

Class	Average Values of IoU
Lung	0.97
Lobe	0.81
Lesion	0.80
Lesion size <2% of the entire lung surface	0.83
Lesion size between 2 and 5% of the entire lung surface	0.81
Lesion size between 5 and 10% of the entire lung surface	0.81
Lesions >10% of the entire lung surface	0.78

## Data Availability

The datasets used and/or analyzed during the current study are available from the corresponding author on reasonable request.

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
