# Peer review of "Training Convolutional Neural Networks to Score Pneumonia in Slaughtered Pigs"

_animals, 2021, doi:10.3390/ani11113290_

Round 1
Reviewer 1 Report
Dear authors,
Thank you for submitting such an interesting paper. The paper documents the development of an AI-based method to classify EP-like lesions at slaughter in finisher pigs. The paper is easy to read and to understand, despite the advanced AI-methods explained. I believe this work to represent an important step for the development of risk-based meat inspection to safeguard public health, but also for the automatization of data collection with the purpose of feeding back information for farmers, helping them to better tackle health and welfare issues on farm and increase production performance.
However, I do have some concerns about the methods described.
I understand it is necessary to proof the concept and I accept this paper to be a great example of how AI can help abattoir-based inspections and assessments. Unfortunately, it is likely that the positive results obtained reflect the methods used, which neglected lungs with artifacts. This introduces bias to the method proposed and should be further considered in the discussion. I disagree with the authors when they claim that “the proportion of lungs with artifacts may not be important” because it is very relevant in some batches and special attention pigs. These batches and pigs are precisely the ones which should be specially assessed and considered in order to safeguard public health, and in order to provide important feedback to the farmer who submitted those pigs for slaughter.
Please see below further comments and suggestions.
Kind regards
Simple summary
L16 – it’s arguable that breeding herds will benefit from scoring slaughter lesions. Maybe remove “entire” at the end of this line?
L18 – such AS those
L19-20 – Once you are continuing the sentence that started with “Artificial intelligence offers interesting opportunities to solve highly repetitive tasks”, finishing it with “as well as to consistently analyze large amounts of data” does not make sense. I suggest changing it to “and to consistently analyse…” instead of “as well as to”.
L22 – remove comma
Abstract
L28 – such AS those
L30 – remove comma
L32 – remove comma
Introduction
L69 – “naturally” instead of “obviously”?
L69-70 – suggestion: … and may or may not leave a scar in the affected area of the lung (“fissures”) which can be detected at slaughter. The complete healing or scarring process will depend on the timing of infection and on the market age of the pigs.
L82 – “combined with the quantification of each lobe” not understandable… I think you mean “combined with a method to account for lobe weights”?
L103 – “on” instead of “about”
L104 – remove “the”
Materials and Methods
L108-110 – “Two slaughterhouses were in Italy (“heavy” pigs, approximately 160 Kgs and 9-10 months of age) and one in Spain (“light” pigs, approximately 90 Kgs and 5 months of age).”
Suggestion: “Two slaughterhouses were located in Italy and one slaughterhouse was located in Spain. The Italian slaughterhouses were processing “heavy” pigs with an approximate slaughter weight of 160 Kgs and an average age of 9-10 months, and the Spanish slaughterhouse was processing “light” pigs with an approximate slaughter weight of 90 Kgs and 5 months of age”.
L111 – are these vets working as official meat sanitary inspectors at slaughter? What kind of training did these vets receive? Have you tested the inter-agreement rate between veterinarians?
L116-117- Out of all lungs, what is the % of lungs with blood or “ripped” that were not included in the study?
Figure 1 – substitute “jugulation” by “exsanguination”
Did you take pictures of the right lungs too? Or does this just take into account the left lung?
Results
L221 – What’s the breakdown left vs right lungs assessed?
Table 1 – space missing in 3rd column - “picturescorrectly”
Space needed between Table 1 and text below.
L252 – please be coherent throughout the text – middle lobe = cranial lobe. Choose one and stick to it. Figure 7 also has “middle lobe”.
Discussion
L320-333 – Overall, severe artifacts may not be important. However, on an individual batch level, these can represent large proportions of the lungs processed. A clear example would be that of a diseased batch or the last pigs of the batch to be sold (typically, slow growing pigs which faced health and welfare impairments). In these cases, which presumably may also represent a threat to public health, the DLBM would fail to identify most lesions. How would the authors overcome this problem?
Not knowing the proportion of lungs with artifacts and not having trained the DLBM on lungs with aspired blood or ripped lungs/lungs with pleurisy represents an important amount of uncertainty.
Author Response
Dear Author,
Please, find attached our reply to your greatly appreciated comments.
Regards,
Giuseppe Marruchella and co-Authors

Reviewer 2 Report
Dear authors,
The manuscript is well written and present good scientific data.
My only concern is about what you detect. Does the system developed differ EP from other causes of lung lesions such as pasteurela app, inlfuenza and etc? If not, should it be used for Pnemonia Index and the introduction section improved?
Best wishes
Author Response
Dear Reviewer,
Please, see the attachment.
Warmest thanks and best regards,
Giuseppe Marruchella & co-Authors

Reviewer 3 Report
Dear authors,
The study entitled „TRAINING CONVOLUTIONAL NEURAL NETWORKS TO SCORE PNEUMONIA IN SLAUGHTERED PIGS“ is an interesting original research with aim to develop an AI-based method capable of recognizing and quantifying enzootic pneumonia-like lesions on digital images, captured from slaughtered pigs under slaughterhouse conditions.
This study has been carried out to train an AI-based system, aiming to automatically score enzootic pneumonia-like lesions on digital images of lungs. The automation of lesion scoring would be extremely helpful to enable a systematic examination of all slaughtered livestock, positively infuencing herd management, animal welfare and profitability.
The present study represents one of the few application of AI-based technologies to detect and quantify lung lesions in slaughtered pigs. In addition, very few papers have yet been published regarding the application of AI to veterinary pathology.
This would allow the systematic collection of data at slaughter, making available an enormous amount of data, useful for better health management of livestock. Using new technologies veterinarians should be able to improve their professional activity. The use of AI reduces the need for labor and saves time and reduces costs.
The training of convolutional neural networks to identify and score pneumonia, on the one hand, and the achievement of trials in large capacity slaughterhouses, on the other, represent the natural pursuance of the present study. As a result, convolutional neural network-based technologies could provide a fast and cheap tool to systematically record lesions in slaughtered pigs, thus supplying an enormous amount of useful data to all stakeholders in the pig industry.
The present investigation further widens the field of application of DLBMs, which now includes another major component of the PRDC. In the next future, AI-based methods could allowing a further leap in quality in the field of food inspection, supporting official veterinarians and optimizing the management of human resources, especially in high-throughput slaughterhouses.
The authors performed appropriate analytical and statistical methods.
I believe this manuscript is useful and can be published in Animals and no additional corrections are needed.
Best regards
Author Response

(The authors gave the same response as above.)

Round 2
Reviewer 1 Report
Dear authors,
Thank you a lot for your revisions. I have no further comments or questions and I recommend this paper to be accepted. Great work!
Kind regards